# Interplay between human STING genotype and bacterial NADase activity regulates inter-individual disease variability

Elin Movert[1], Jaume Salgado Bolarin[1], Christine Valfridsson[1], Jorge Velarde[2], Steinar Skrede [3,4], Michael Nekludov[5], Ole Hyldegaard[6,7], Per Arnell[8], Mattias Svensson [9], Anna Norrby-Teglund[9], Kyu Hong Cho[10], Eran Elhaik [1], Michael R. Wessels [2], Lars Råberg [1] & Fredric Carlsson [1] ✉

Variability in disease severity caused by a microbial pathogen is impacted by each infection representing a unique combination of host and pathogen genomes. Here, we show that the outcome of invasive *Streptococcus pyogenes* infection is regulated by an interplay between human STING genotype and bacterial NADase activity. *S. pyogenes*-derived c-di-AMP diffuses via streptolysin O pores into macrophages where it activates STING and the ensuing type I IFN response. However, the enzymatic activity of the NADase variants expressed by invasive strains suppresses STING-mediated type I IFN production. Analysis of patients with necrotizing *S. pyogenes* soft tissue infection indicates that a STING genotype associated with reduced c-di-AMP-binding capacity combined with high bacterial NADase activity promotes a 'perfect storm' manifested in poor outcome, whereas proficient and uninhibited STING-mediated type I IFN production correlates with protection against host-detrimental inflammation. These results reveal an immune-regulating function for bacterial NADase and provide insight regarding the host-pathogen genotype interplay underlying invasive infection and interindividual disease variability.

Since the mid-1980s there has been a surge of invasive *S. pyogenes* infections, including necrotizing soft tissue infection (NSTI), a disease characterized by excessive and tissue-degrading inflammation associated with high fatality rates[1–3]. However, the outcome of these invasive infections varies considerably among patients, for unknown reasons. The origin of the epidemic is linked to a horizontal gene transfer event whereby serotype M1 *S. pyogenes* acquired an allele-variant of the *nga* gene encoding an enzymatically active NAD-glycohydrolase (NADase) as well as a promoter that drives high-level expression of the operon encoding both NADase and the pore-forming toxin streptolysin O (SLO)[2–4]. Secreted NADase enters the host cell cytosol via SLO-dependent so-called cytolysin-mediated translocation (CMT) across the plasma membrane[5,6]. Within cells, active NADase depletes NAD by cleaving the molecule into ADP-ribose and

[1]Department of Biology, Lund University, Sölvegatan 35, 223 62 Lund, Sweden. [2]Division of Infectious Diseases, Boston Children's Hospital, and Department of Pediatrics, Harvard Medical School, Boston, USA. [3]Department of Medicine, Haukeland University Hospital, Bergen, Norway. [4]Department of Clinical Science, University of Bergen, Bergen, Norway. [5]Department of Anaesthesia, Surgical Services and Intensive Care, Karolinska University Hospital, Stockholm, Sweden. [6]Department of Anaesthesia, Head and Orthopedic Center, Rigshospitalet, Copenhagen, Denmark. [7]Department of Clinical Medicine, University of Copenhagen, Copenhagen, Denmark. [8]Department of Anaesthesia and Intensive Care, Sahlgrenska University Hospital, Gothenburg, Sweden. [9]Centre for Infectious Medicine, Department of Medicine Huddinge, Karolinska Institutet, Karolinska University Hospital, Stockholm, Sweden. [10]Department of Biology, Indiana State University, Terre Haute, USA. ✉e-mail: fredric.carlsson@biol.lu.se

nicotinamide[7], resulting in ATP deprivation and ultimately necrotic cell death[7,8]. Consistent with this function, studies in animal infection models have established enzymatically active NADase as a major virulence factor[2,3,9]. Whilst the toxic effect of NAD degradation may promote inflammation and invasiveness, NAD is a conserved molecule suggesting that the NADase-NAD interaction alone is unlikely to account for the variability in disease outcome observed among patients with invasive conditions. Thus, we hypothesized that NADase interacts with a more dynamic host process – influenced by genetic polymorphisms – to regulate the inflammatory response.

Studies of lethal *S. pyogenes* soft tissue infection in mice have demonstrated that type I interferon (IFN) signaling promotes host survival by suppressing host-detrimental inflammation[10,11]. In macrophages, *S. pyogenes* induces type I IFN production via the evolutionarily conserved stimulator of interferon genes (STING) cytosolic surveillance pathway[11–13]. STING activation may occur as a result of DNA leakage into the host cell cytosol, where the misplaced DNA is sensed by the cyclic GMP-AMP synthase (cGAS) to produce 2′, 3′-cGMP-AMP (cGAMP), an endogenous cyclic dinucleotide that binds to and activates STING[14]. However, we recently found that *S. pyogenes* activates STING independently of cGAS and cytosolic sensing of DNA[12], suggesting that STING might be directly activated by *S. pyogenes*-derived c-di-AMP. As human STING exhibits single nucleotide polymorphisms (SNPs) affecting its ability to interact with c-di-AMP[15–19], we here considered type I IFN induction a candidate process for NADase-dependent host-pathogen genotype interactions.

## Results

### *S. pyogenes*-derived c-di-AMP diffuses into the macrophage cytosol via SLO pores to activate STING directly

The *S. pyogenes* c-di-AMP synthase (DacA) produces c-di-AMP that can be secreted into the extracellular space[20]. The metabolism of c-di-AMP further includes two phosphodiesterases, which linearize (GdpP) and cleave (Pde2) it into two AMP molecules in a sequential two-step process (Fig. 1a)[20]. To investigate if bacteria-derived c-di-AMP activates STING directly we employed isogenic deletion mutants of an *S. pyogenes* M14 (HSC5) strain affected in the different steps of c-di-AMP metabolism[20]. Absence of the synthase ablated secretion of c-di-AMP whereas bacteria unable to linearize c-di-AMP due to lack of GdpP secreted excess amounts (Fig. 1b). Lack of Pde2, which acts downstream of linearization, had a minor but significant effect and a double mutant lacking both phosphodiesterases secreted drastically increased levels of c-di-AMP (Fig. 1b). While deletion of DacA caused a prolonged lag-phase in broth cultures, all mutants exhibited similar growth rates as wild type in exponential phase when they were harvested for macrophage infections (Supplementary Fig. 1). The amount of secreted c-di-AMP correlated with *ifnβ* transcription in infected macrophages and no expression was detected in ΔdacA infection (Fig. 1c), indicating that bacterial c-di-AMP is required to activate STING. Infection of STING-deficient cells demonstrated that the increased induction of type I IFN by mutants secreting excess amounts of c-di-AMP was completely dependent on STING (Supplementary Fig. 2a) and did not relate to differential cell death (Supplementary Fig. 2b). In contrast to IFNβ, the production of TNFα in *S. pyogenes*-infected macrophages requires MyD88-signaling[12] and was unaffected by c-di-AMP (Fig. 1d), suggesting a selective effect of bacteria-derived c-di-AMP on type I IFN production. Analysis at the protein level confirmed a direct role for secreted c-di-AMP in activating the STING pathway (Fig. 1e). Production of the neutrophil attractant CXCL1 in infected cells was inhibited by type I IFN signaling as indicated by the increased secretion from type I IFN receptor-deficient macrophages (Supplementary Fig. 2c) and, accordingly, it demonstrated an essentially reciprocal pattern of secretion as compared to IFNβ upon infection with our mutants (Fig. 1f).

To test if *S. pyogenes*-derived c-di-AMP is sufficient to activate STING in macrophages, we devised a bacteria-free system where digitonin was used to transiently permeabilize the cell membrane to allow access to the intracellular compartment. Both pure c-di-AMP, which was used as a control, and sterilized culture supernatant (culture filtrate; CF) from ΔgdpPΔpde2 bacteria – secreting high amounts of c-di-AMP (Fig. 1b) – were sufficient to induce *ifnβ* expression in permeabilized macrophages (Fig. 1g). Macrophages not treated with digitonin failed to respond (Fig. 1g), consistent with the need for c-di-AMP to enter the cytosol to activate STING. Pretreatment of c-di-AMP and culture filtrate with phosphodiesterase I (PD) to digest c-di-AMP abrogated the signal (Fig. 1g), pinpointing a critical role for c-di-AMP in the bacterial culture filtrate.

*S. pyogenes* produces SLO, a cholesterol-dependent pore-forming toxin that inserts into and permeabilizes host cell plasma membranes[6]. To test if bacterial c-di-AMP enters macrophages via diffusion through SLO pores we took advantage of an *S. pyogenes* M3 strain and its isogenic SLO-deficient mutant (Δslo)[5]. Wild type and Δslo bacteria secreted similar amounts of c-di-AMP (Fig. 1h). However, analysis at the mRNA (Fig. 1i) and protein (Fig. 1j) levels showed that SLO-deficient *S. pyogenes* was unable to induce IFNβ production in infected macrophages, whereas it generated a normal TNFα response (Fig. 1k). In agreement with this finding, SLO was required to induce type I IFN receptor signaling as measured by STAT1 phosphorylation (Fig. 1l and Source Data in Supplementary Information). Addition of digitonin to Δslo infected macrophages restored type I IFN production (Fig. 1m), supporting the interpretation that SLO pores are required to allow passage of bacterial c-di-AMP into the host cell cytosol.

### NADase inhibits *S. pyogenes*-induced type I IFN production in infected macrophages

To probe the broader hypothesis that active NADase promotes host-detrimental inflammation by suppressing STING-mediated type I IFN production we infected macrophages with SF370, a so-called pre-epidemic (*i.e.* encoding an enzymatically inactive NADase allele-variant) M1 strain isolated before the horizontal gene transfer event in the 1980s[2], as well as 5448 and 854, two epidemic (encoding enzymatically active NADase allele-variants) M1 isolates[2]. Strikingly, unlike SF370 the epidemic strains essentially did not induce any type I IFN production (Fig. 2a, b). Infection with the epidemic strains also gave rise to a ~3-fold reduced TNFα response as compared to SF370 (Fig. 2c).

The acquired NADase-encoding operon in epidemic strains also encodes SLO and a functional cytosolic immunity factor (IFS) that tolerizes *S. pyogenes* to NADase by inhibiting its activity[21]. Secreted NADase and SLO interact to stabilize each other in the extracellular environment[22]. SLO then targets the complex to host cell membranes and is required for cytosolic translocation of NADase via CMT[5,6] – a process that is independent of SLO pore formation[23,24]. To determine the causal relationship, if any, between the enzymatic activity of NADase and the effects on cytokine production we infected macrophages with 854 wild type and isogenic mutants affected in NADase and/or SLO function[22]. The exceptionally low level of IFNβ secretion observed in 854 wild-type infection was not further reduced by deletion of SLO (Fig. 2d), a mutation that denies access to the macrophage cytosol for both bacteria-derived c-di-AMP and NADase. IFNβ was similarly unaffected by the introduction of an amino acid substitution (sloY255A) rendering SLO incapable of forming pores while maintaining ability to interact with NADase and perform CMT (Fig. 2d)[22]. Lack of NADase (Δnga) destabilizes secreted SLO[22], leading to reduced diffusion of c-di-AMP into macrophages, which likely explains why lack of NADase did not affect IFNβ output as compared to wild-type infection (Fig. 2d). Introduction of the ngaG330D amino acid substitution renders NADase enzymatically inactive while maintaining its ability to stabilize SLO[22], allowing proficient diffusion of c-di-AMP via SLO pores while translocating the enzymatically inactive NADase. Importantly,

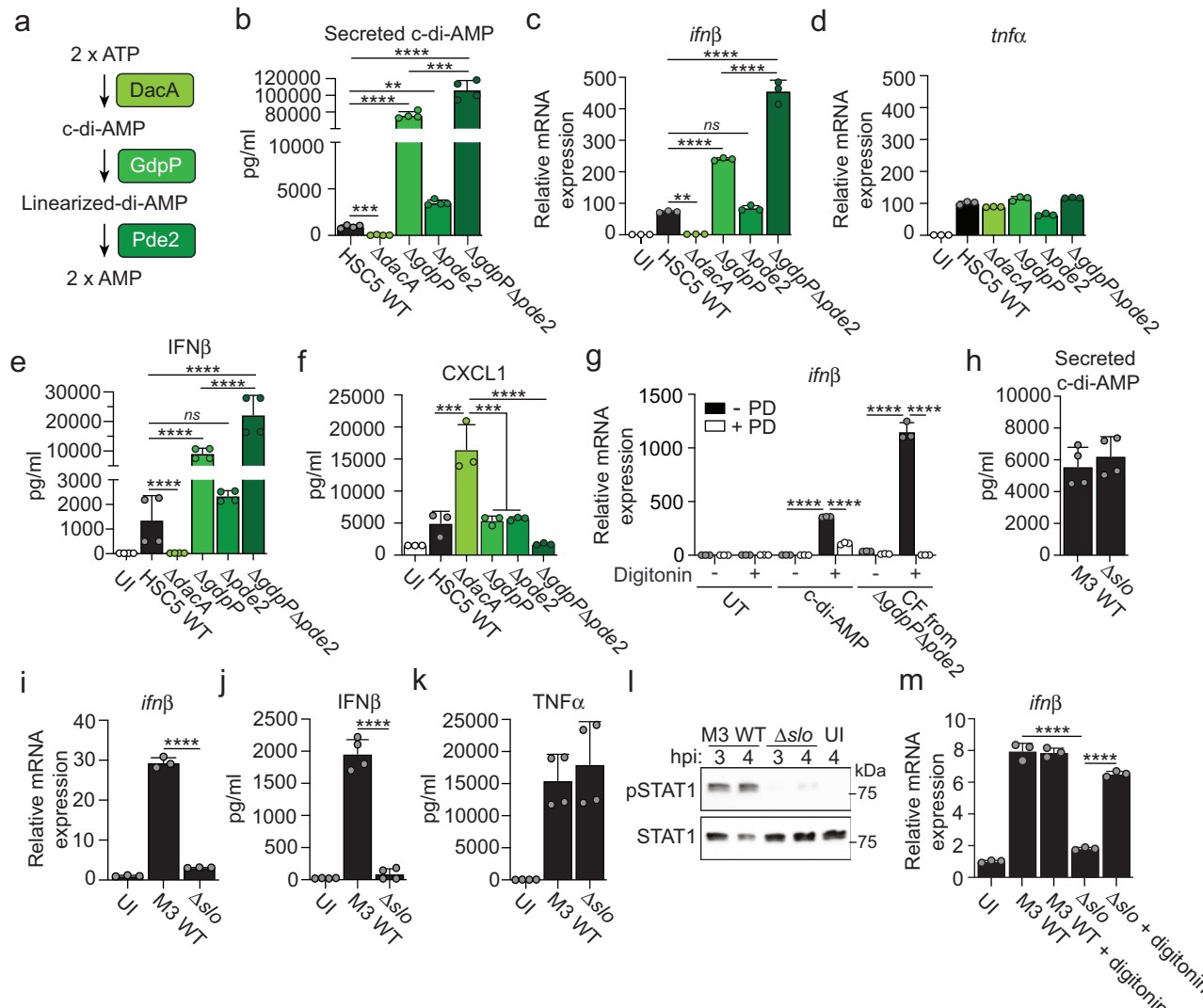

**Fig. 1 | Bacteria-derived c-di-AMP access the macrophage cytosol via SLO pores and activates STING directly. a** Schematic overview of the *S. pyogenes* enzymes involved in the synthesis (DacA) and degradation (GdpP and Pde2) of c-di-AMP. **b** Analysis of c-di-AMP concentration in culture supernatants (i.e. secreted c-di-AMP) from *S. pyogenes* M14 wild type (HSC5 WT) and deletion mutants, as indicated. **c, d** Gene expression analysis of *ifnβ* (**c**) and *tnfα* (**d**) in C57Bl/6 macrophages infected as indicated, at 4 h post infection (hpi). Uninfected (UI) macrophages were analyzed as control. **e, f** ELISA-based analysis of secreted IFNβ (**e**) and CXCL1 (**f**) from macrophages infected, as indicated, at 20 hpi. **g** *ifnβ* expression in transiently permeabilized macrophages (±10 μg/ml digitonin) treated with pure c-di-AMP (7.5 μM) or sterilized culture supernatant (culture filtrate; CF) from ΔgdpPΔpde2 bacteria, with or without phosphodiesterase I (±PD; 20 U/ml) pre-incubation.

Untreated (UT) macrophages were analyzed as control. **h** Analysis of c-di-AMP secreted from *S. pyogenes* M3 WT and its isogenic SLO deletion mutant (Δ*slo*). **i** Gene expression analysis of *ifnβ* in macrophages infected, or UI, as indicated, at 4 hpi. **j, k** ELISA-based analysis of secreted IFNβ (**j**) and TNFα (**k**) at 20 hpi, as indicated. **l** Western blot analysis, at the indicated hpi, of activated (i.e., phosphorylated; pSTAT1) and total amount of STAT1 in macrophages infected with M3 WT or Δ*slo*. Shown is one experiment representative of two. Molecular size (kDa), as indicated. **m** *ifnβ* expression at 4 hpi in M3 WT or Δ*slo* infected macrophages ± digitonin treatment. Results for gene expression analysis (mean and SD; *n* = 3), secreted c-di-AMP (mean and SD; *n* = 3), and ELISA analysis (mean and SD; *n* = 3 in **f** and *n* = 4 in **e, j, k**) representative of at least three independent experiments. One-way ANOVA with Dunnett's test. *$p < 0.05$, **$p < 0.01$, ***$p < 0.001$, ****$p < 0.0001$.

the *nga*G330D substitution drastically increased secretion of IFNβ from infected macrophages (Fig. 2d), demonstrating that NADase activity inhibits STING-mediated type I IFN production. Analysis of STING-deficient macrophages confirmed that the elevated production of type I IFN in *nga*G330D infection was fully dependent on STING (Supplementary Fig. 3). The *slo*Y255A*nga*G330D double mutant translocates the enzymatically dead NADase into macrophages via CMT but is unable to form SLO pores[22], and was unable to induce type I IFN production (Fig. 2d). Thus, consistent with the M3 system (Fig. 1h−m), our results with the *nga*G330D and *slo*Y255A*nga*G330D amino acid substitution mutants in the 854 strain background (Fig. 2d) indicate that SLO pore formation is required for the diffusion of bacteria-derived c-di-AMP into the host cell cytosol to activate STING.

The TNFα output from infected cells was not specifically regulated by NADase activity (Fig. 2e), indicating that the enzymatic activity of NADase selectively inhibits production of type I IFN.

Analysis of *ifnβ* gene expression at 4 h post infection similarly demonstrated a critical role for enzymatically active NADase in suppressing the type I IFN response (Fig. 2f), indicating regulation at the transcriptional level. At this early time point we observed limited but significant *ifnβ* expression in Δ*nga* infection (Fig. 2f) − possibly due to residual SLO[22] leading to a limited diffusion of c-di-AMP into macrophages − which was not reflected at the protein level at 20 h post infection (Fig. 2d). The ability of the bacterial strains to suppress STING-mediated type I IFN production was not a result of differential secretion of c-di-AMP (Fig. 2g) or cell death (Supplementary Fig. 4) but

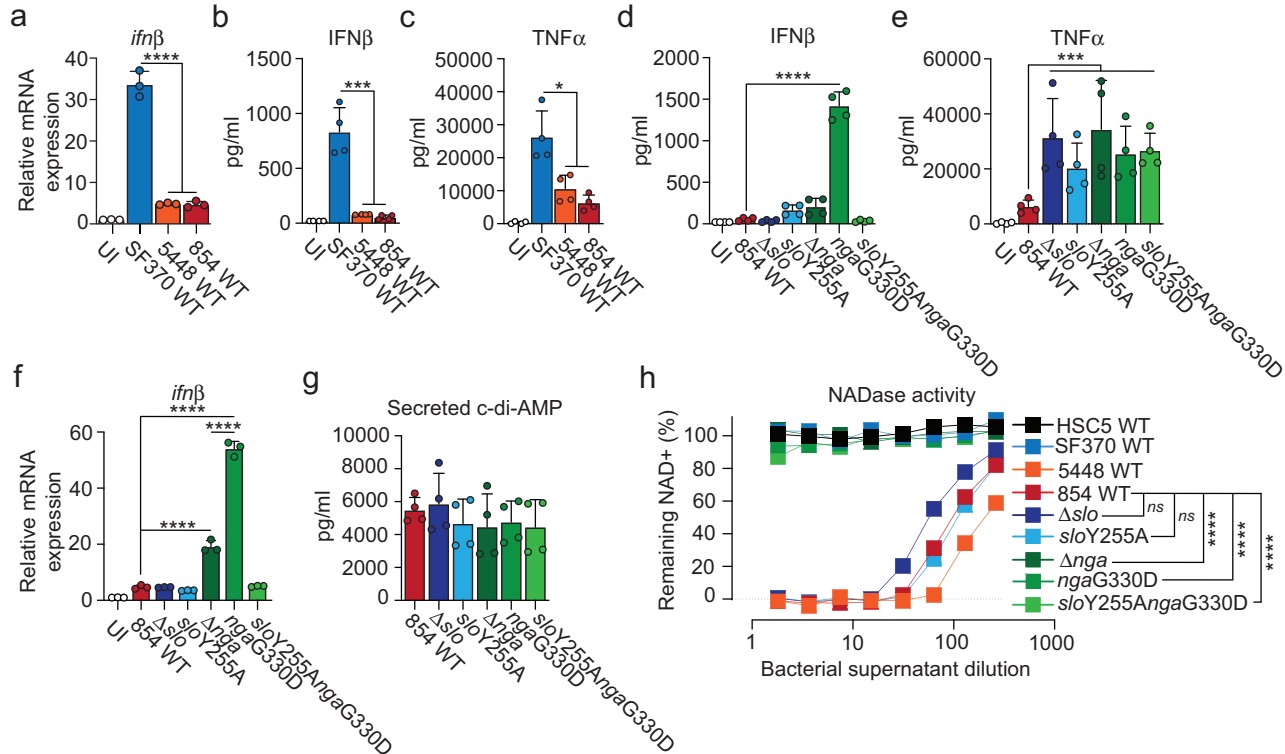

**Fig. 2 | Enzymatically active NADase inhibits type I IFN production in infected macrophages. a** Gene expression analysis of *ifnβ* in C57Bl/6 macrophages at 4 hpi with M1 strains SF370, 5448 or 854. Uninfected cells (UI) were analyzed as control. **b, c** ELISA-based analysis of secreted IFNβ (**b**) and TNFα (**c**) in similarly infected macrophages at 20 hpi. **d, e** Analysis of secreted IFNβ (**d**) and TNFα (**e**) from macrophages infected with 854 WT and thereof derived isogenic mutants, as indicated, at 20 hpi. **f** RTqPCR-based analysis of *ifnβ* expression at 4 hpi. **g** Secreted levels of c-di-AMP from 854 WT and isogenic mutants, as indicated. Results for gene expression analysis (mean and SD; $n = 3$), secreted c-di-AMP (mean and SD; $n = 4$) and cytokine secretion analyses (mean and SD; $n = 4$) representative of at least three independent experiments. One-way ANOVA with Dunnett's test. **h** Enzymatic activity of NADase in bacterial supernatants diluted as indicated was assessed by NAD degradation. Results based on at least three independent experiments. Two-way ANOVA with Tukey's test were used for multiple comparisons. *$p < 0.05$, **$p < 0.01$, ***$p < 0.001$, ****$p < 0.0001$.

related specifically to their ability to translocate an enzymatically active NADase (Fig. 2d–f, h).

## NADase acts intracellularly to inhibit *ifnβ* transcription via a mechanism that is dependent on ATP depletion

To understand the mechanism by which NADase suppresses induction of type I IFN we stimulated digitonin-treated macrophages with agonists that drive *ifnβ* expression via different signaling pathways and tested the ability of purified enzymatically active and inactive (carrying the G330D substitution) NADase proteins to inhibit the response. c-di-AMP, c-di-GMP and double-stranded DNA induce type I IFN production via the STING pathway[14]. Stimulation with c-di-AMP and c-di-GMP, as well as transfection with double-stranded DNA, all induced *ifnβ* transcription that was significantly suppressed by the active but not by the inactive protein (Fig. 3a). Type I IFN induction by the TLR3 agonist poly(I:C) was also specifically inhibited by the enzymatically active NADase (Fig. 3a), indicating that NADase acts downstream of STING activation and that its mode-of-action suppresses *ifnβ* expression not only when induced via the STING pathway.

Unlike cyclic dinucleotides, the TLR4 agonist LPS does not require membrane permeabilization to induce type I IFN production. Purified NADase effectively inhibited LPS-induced *ifnβ* expression in digitonin permeabilized macrophages but had only a minor effect in non-permeabilized cells (Fig. 3b), indicating that NADase acts intracellularly to inhibit the type I IFN response. Supplementation with exogenous ATP completely reversed the ability of NADase to inhibit c-di-AMP-induced transcription of *ifnβ* (Fig. 3c). ATP alone did not induce *ifnβ* expression in macrophages (Supplementary Fig. 5a) nor

did it inhibit the enzymatic activity of NADase (Supplementary Fig. 5b, c), suggesting that NADase blocks *ifnβ* expression via a mechanism that depends on reduced availability of cellular ATP. Direct analysis of ATP content confirmed that the enzymatic activity of NADase rapidly reduced cellular ATP (Fig. 3d). NADase did not promote lactate dehydrogenase (LDH) release during the time frame of these experiments (Fig. 3e), suggesting that its effect on type I IFN production is not a consequence of cell death.

Collectively our results indicate that SLO pores allow diffusion of *S. pyogenes*-derived c-di-AMP into the macrophage cytosol where it is both required and sufficient to activate STING (Fig. 3f). However, concomitant translocation of the enzymatically active NADase expressed by epidemic strains suppresses type I IFN production at the transcriptional level via a mechanism that is dependent on ATP-exhaustion but unrelated to cell death (Fig. 3f).

## Interplay between human STING genotype and bacterial NADase activity regulates disease outcome in NSTI patients

Human STING exhibits polymorphisms that affect stimulation by endogenous cGAMP and bacterial c-di-AMP. The STING-R71H-G230A-R293Q variant (STING-HAQ; here analyzed by SNP rs78233829, G230A) has an allele frequency that ranges from 9.1 to 44% globally and exhibits impaired responsiveness to both cGAMP and c-di-AMP, whereas the STING-R232H variant (SNP rs1131769) occurs at frequencies of 5.9–24% and is selectively unresponsive to c-di-AMP[15–19]. Thus, both of these minor alleles encode proteins that respond poorly to c-di-AMP. Murine and human cell studies suggest a gene dose effect for both variants, where heterozygotes have an intermediate phenotype[18,19].

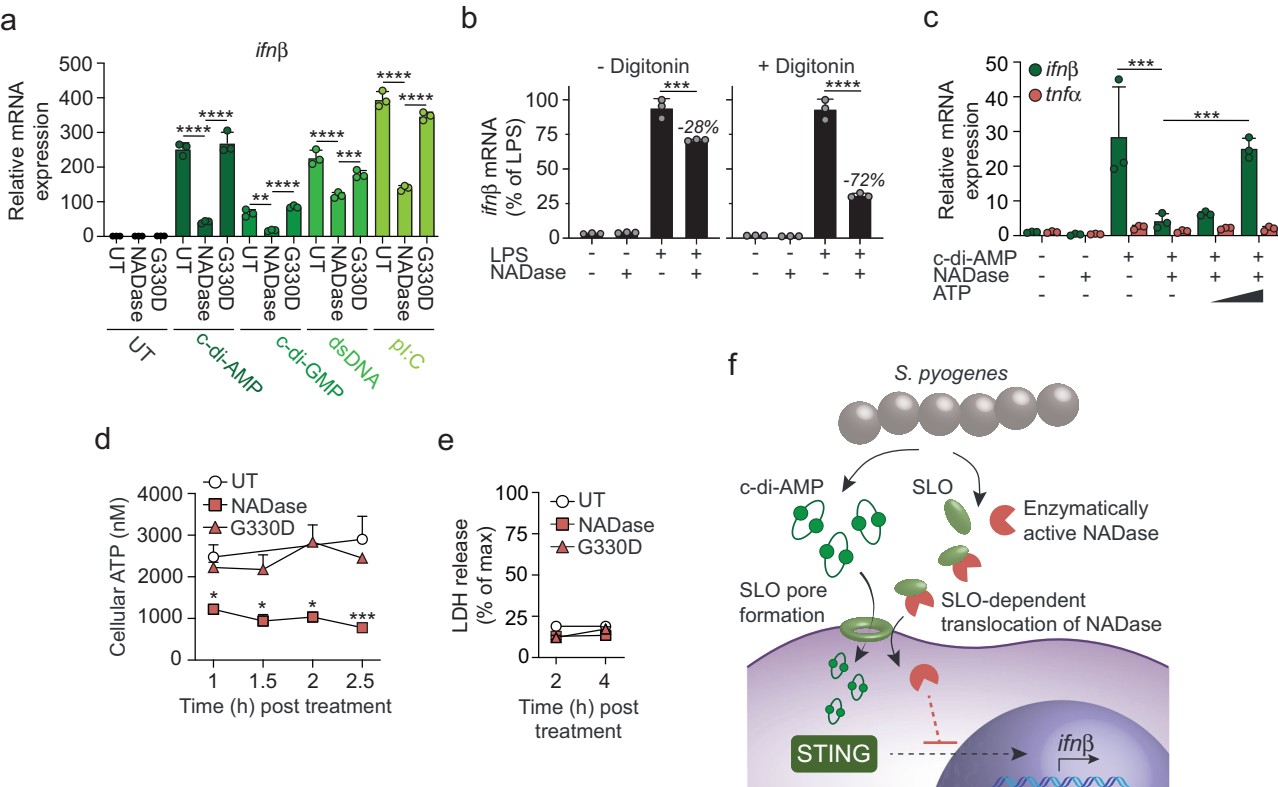

**Fig. 3 | NADase acts intracellularly to inhibit *ifnβ* transcription via an ATP-depletion dependent mechanism. a** RTqPCR analysis of *ifnβ* expression in transiently permeabilized (10 µg/ml digitonin) C57Bl/6 macrophages treated with 0.5 µM purified enzymatically active (NADase) or inactive (G330D) recombinant NADase, or untreated (UT), in combination with c-di-AMP (7.5 µM), c-di-GMP (7.5 µM), dsDNA (2.5 ng/ml), pI:C (2.5 µg/ml) or UT, as indicated. Analysis was performed 2,5 h post treatment. **b** LPS (2.5 µg/ml)-induced *ifnβ* expression in permeabilized (+digitonin) or non-permeabilized (- digitonin) macrophages treated with enzymatically active NADase (0.5 µM), as indicated. *ifnβ* expression is presented as % of LPS alone. **c** Expression of *ifnβ* and *tnfα* in permeabilized macrophages treated with enzymatically active NADase (0.5 µM) in combination with c-di-AMP (7.5 µM), and with the addition of titrated amounts of ATP (0.1 mM and 1 mM),

as indicated. **d** Kinetic analysis of cellular ATP concentration in transiently permeabilized (+digitonin) macrophages treated with NADase, G330D, or untreated (UT), as indicated. **e** Kinetic analysis of LDH-release (as a measure of cytotoxicity) in cells treated as in **d** above. All results (mean and SD; $n = 3$) are representative of three independent experiments. **a**–**c** One-way ANOVA with Dunnett's test, **d**, **e** two-way ANOVA with Tukey's test. *$p < 0.05$, **$p < 0.01$, ***$p < 0.001$, ****$p < 0.0001$. **f** Schematic representation of our results. SLO pores allow diffusion of *S. pyogenes*-derived c-di-AMP into the macrophage cytosol where it activates STING to drive *ifnβ* expression. However, simultaneous CMT of the enzymatically active NADase expressed by epidemic strains suppresses type I IFN production at the transcriptional level via a mechanism that is dependent on ATP depletion but unrelated to cell death.

Interestingly, the present-day regional differences in allele frequencies might reflect local adaptations that began after the human-chimpanzee split and intensified during the Neolithic revolution (Supplementary Fig. 6).

To determine if the combination of high bacterial NADase activity with poor ability of STING to respond to c-di-AMP manifests in poor disease outcome, we examined 73 NSTI patients admitted at five different hospitals in Scandinavia (Fig. 4a)[25]. For 53 of the patients we obtained the corresponding bacterial isolates, which covered 13 different *emm*-types (Supplementary Fig. 7). All strains encoded an enzymatically active NADase variant as indicated by sequence and quantitative functional analyses (Supplementary Figs. 7, 8). Sequencing of the STING gene (*STING1* a.k.a. *TMEM173*) indicated similar allele frequencies among patients as for the general Swedish population (Fig. 4b), suggesting that STING genotype does not affect susceptibility to NSTI. Of the patients, 33 were homozygous for the major allele whereas 40 had at least one copy of a minor allele (Fig. 4c). No differences in age, sex, BMI, or comorbidities were observed between these two groups (Supplementary Fig. 9). Strikingly, while the mortality was similar in both groups (2 fatal cases per group), only 1 out of 27 major allele patients (3.7%) required amputation of the infected extremity as compared to 7 out of 34 minor allele patients (20.6%; Fig. 4d), indicating a critical host-protective role for STING

responsiveness to bacterial c-di-AMP. STING genotype alone did not similarly manifest in septic shock (Fig. 4e), suggesting a role for STING mainly within the inflamed tissue. However, individuals infected with strains expressing relatively higher NADase activity were more likely to develop this systemic inflammatory condition (Fig. 4f). The amount of secreted c-di-AMP did not similarly correlate with septic shock (Supplementary Fig. 10). Though the expression levels of *slo* and *nga* co-vary[3], these findings identify NADase activity, and not c-di-AMP secretion, as the key bacterial variable across different strains and *emm*-types. Moreover, the positive correlation between bacterial NADase activity and septic shock was significant only in patients homozygous for the major allele (Fig. 4g), and not in patients with ≥1 minor allele (Fig. 4h), suggesting a requirement for NADase-mediated suppression of STING signaling – for the development of septic shock – primarily when STING can be proficiently activated by *S. pyogenes*-derived c-di-AMP.

## Discussion

It is well established that different strains of a pathogenic microorganism can exhibit different degrees of virulence. However, disease severity caused by a single strain may also vary between immunocompetent host individuals, a phenomenon of fundamental importance likely impacted by each infection representing a unique

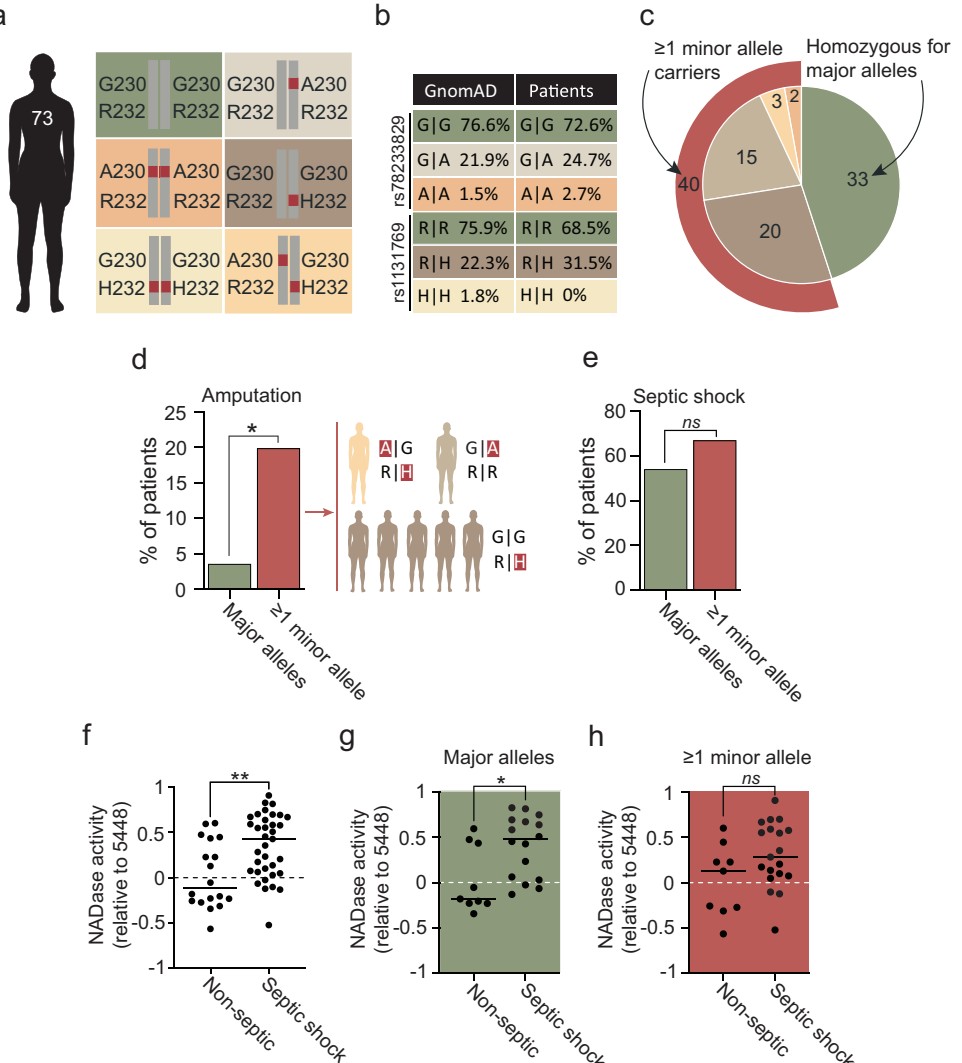

**Fig. 4 | Functional interplay between human STING genotype and bacterial NADase activity regulates disease outcome in NSTI patients. a** Overview of the STING alleles and combinations analyzed for the 73 NSTI patients, amino acid substitutions G230A and R232H, representing the minor alleles, are indicated in red. **b** Allele frequency analysis of the two SNPs rs78233829 and rs1131769 corresponding to amino acid G or A at position 230, and R or H at position 232, respectively. The patient cohort was compared to ~13,000 Swedish individuals in the Genome Aggregation Database (GnomAD), representing the allele frequencies within the general Swedish population (Fisher's exact test: rs87233829: $p = 0.6$; rs1131769: $p = 0.13$). **c** Pie chart summary of the STING genotypes of the 73 NSTI patients. Color coding relates to the amino acid substitutions indicated in **a**. **d** Frequencies of major and minor allele carriers (as defined in **c**), respectively, that

required amputation of the infected limb. Specific STING genotypes of amputated minor allele carriers are indicated to the right. This analysis is based on 61 NSTI patients with infection localized to extremities, including 27 homozygous major allele carriers and 34 minor allele carriers (generalized linear model: genotype: $\chi^2 = 4.48$, df = 1, $p = 0.034$). **e** Frequencies of major and minor allele carriers (as defined in **c**), respectively, that developed septic shock (genotype: $p > 0.22$). **f** Relative NADase activity in overnight cultures of the 53 *S. pyogenes* NSTI patient isolates grouped based on whether or not the patients developed septic shock. **g, h** Same analysis as in **f** above, but *S. pyogenes* isolates where further divided based on whether the patients were homozygote major allele carriers (**g**) or minor allele carriers (**h**). Results (**f**–**h**) are based on three independent experiments. Two-sided unpaired *t*-test. *$p < 0.05$, **$p < 0.01$, ***$p < 0.001$, ****$p < 0.0001$.

combination of microbial and host genomes[26]. Still, little is known about the host-pathogen interactions and the evolution of host and pathogen genotype interplay that underlie such variability. In this context *S. pyogenes*, which is responsible for ~700 million cases of disease annually[1], is of particular interest since it evolves exclusively in humans and is able to cause a spectrum of diseases ranging from superficial pharyngitis and impetigo to invasive conditions such as septic shock and necrotizing soft tissue infection[1]. In its invasive manifestations the infection produces an excessive inflammatory response that can result in severe tissue damage and life-threatening disease[1–3]. We identify a new function for the enzymatic activity of bacterial NADase in suppressing c-di-AMP-induced and STING-mediated type I IFN production, a function that contributes to the

increased virulence of epidemic *S. pyogenes* strains in humans. Moreover, we uncover a dynamic host-pathogen interplay between human STING genotype and bacterial NADase activity that regulates disease outcome. The STING-HAQ or STING-R232H variants – exhibiting reduced c-di-AMP-binding capacity[15–19] – and bacterial strains expressing high NADase activity promote poor outcome, whereas proficient STING-mediated type I IFN production conversely protects against host-detrimental inflammation. While our bacterial mutants were not complemented to formally exclude contributions from potential secondary mutations[27], interpretations based on macrophage infection experiments were validated by biochemical approaches and supported by clinical data. It will be important to fully elucidate the mechanism by which NADase suppresses type I IFN production.

Our results indicate that SLO pore formation is required for *S. pyogenes*-derived c-di-AMP to gain access to the host cell cytosol, where it is both required and sufficient to activate STING and the ensuing type I IFN response. Because SLO-dependent translocation is facilitated by the cellular proximity afforded by M protein-mediated adhesion[6], these findings also provide a potential explanation to the previously described role of the streptococcal M protein in type I IFN induction[12].

All clinical strains in our study encoded an enzymatically active NADase, but the positive correlation between NADase activity and septic shock was discernible only in major allele patients, implying less need for the enzyme in minor allele patients with inherently poor response to c-di-AMP. The frequencies of STING polymorphisms, particularly the STING-HAQ variant, differ between human populations[17], a feature that we find emerged since the agricultural revolution and that, in light of our results, might explain the apparently non-uniform association between bacterial NADase and invasive infections in present-day populations. Indeed, two studies in the US have found an association between NADase-active *S. pyogenes* strains and invasive disease[28,29] whereas studies performed on Aboriginal Australian[30] and on worldwide[31] populations, as well as a study in Portugal[32], did not. It will be of interest to investigate if differences in STING allele frequencies between populations might explain this ambiguous situation regarding the link between bacterial NADase activity and invasive manifestations in humans, and to explore the herein described NADase-STING axis for diagnostic purposes and personalized treatment strategies.

## Methods

### Ethical statement
All animal care and use adhered to the Swedish animal welfare laws, and to the guidelines set by the Swedish Department of Agriculture (Act 1988:534). Studies entailing mice were approved by the Malmö/Lund Ethical Board for Animal Research (permit number 5.8.18-07342/2017 and 5.8.18-08454/2020). Mice were housed in disposable cages (2–5 animals per cage; Innovive) with environmental enrichment and free access to food and water. Animals experienced a 12 h day/night cycle and were inspected daily. The INFECT study is a multicenter, prospective observational cohort study registered at ClinicalTrials.gov (NCT01790698). Design, study sites, inclusion criteria, and ethics of the cohort are previously described[33]. Herein described work was approved by the Danish Ethical Committee (1211709), the Swedish Ethical Committee (Dnr: 930-12), and the Regional Committee for Ethics in Medical Research (2012/2227/REK VEST) in Western Norway.

### Bacterial strains
All reagents, kits, and strains used in this study are listed and referenced in Supplementary Table 1. *S. pyogenes* was cultured overnight in Todd-Hewitt broth (Difco Laboratories) supplemented with 0.2% yeast extract (THY) at 37 °C + 5% $CO_2$. Overnight cultures were reinoculated 1:20 in fresh THY and grown statically at 37 °C + 5% $CO_2$.

### Macrophages
Bone marrow-derived macrophages from wild type C57Bl/6JRj, *Ifnar1*[-/-] (IFNAR-KO) or *Tmem173*[gt] (STING-KO) were prepared as previously described[12]. In brief, extracted bone marrow cells were cultured for 7 days in R15 medium (RPMI-1640; Invitrogen) supplemented with 1% L-glutamine, 10 mM Hepes (Sigma), 50 U/ml penicillin G (Gibco), 50 µg/ml streptomycin (Gibco), 10% (vol/vol) heat-inactivated fetal calf serum (Sigma; endotoxin <0.2 EU/ml), and 15% (vol/vol) M-CSF-containing supernatant from 3T3-CSF cells. Wild type and IFNAR-KO mice were bred and maintained at the Biology Department animal facility. Bone marrow from STING-KO animals was isolated and kindly provided by Russell Vance (UC Berkeley, USA). For infection or treatment experiments (see below), $3.75 \times 10^5$ macrophages were seeded into wells of 24-well non-tissue culture treated plates (Corning) and a final volume of 250 µl of suspended bacteria, culture filtrate or treatment mixes was added, as indicated in figure legends.

### Recombinant *S. pyogenes* NADase
NADase and the enzymatically inactive G330D amino acid substitution variant of the protein were cloned and expressed, using IPTG induction, in BL21 (DE3) *Escherichia coli* (Invitrogen), and purified as previously described in detail[34].

### Analyses of bacterial NADase activity and c-di-AMP secretion
To assess the enzymatic activity of NADase, bacterial supernatants were harvested either from exponential phase bacterial cultures or from overnight cultures, and analyzed essentially as previously described[5]. Briefly, supernatants were serially diluted in THY in a 96-well plate and a final concentration of 0.67 mM NAD (N-7004, Sigma) was added at a volume of 1:1. The plate was incubated at 37 °C for 3 h, unless otherwise indicated, before a final concentration of 2 M NaOH was added to develop the reactions. The reactions were developed at room temperature for 30 min under light-protected conditions. Analysis was performed using the SpektraMAX i3x fluorimeter (Molecular Devices) with excitation/emission wavelengths of 370/460 nm. The epidemic reference strain 5448 was included in all separate assays to allow comparative analysis of NADase activity for each of the 53 *S. pyogenes* isolates from NSTI patients. For each experiment, data was normalized to the dilution value corresponding to 50% of remaining NAD for 5448, which then by definition obtained a log-transformed value of zero.

For analysis of secreted c-di-AMP, bacteria were grown to exponential phase in chemically defined medium (CDM)[35]. Supernatants were collected and sterilized by filtration (0.25 µm) before measuring c-di-AMP using the c-di-AMP ELISA Kit (Cayman), according to the manufacturer's instructions.

### Macrophage infections
Infections with *S. pyogenes* were performed essentially as described previously[12]. Briefly, cultures of exponential phase bacteria ($OD_{600}$ ˜ 0.6) were washed twice with DPBS (Invitrogen) and resuspended to the appropriate concentration in serum-free Opti-MEM® (Invitrogen). Macrophages seeded to non-tissue culture-treated plates (Corning) were infected at a multiplicity of infection (MOI) of 20 and incubated at 37 °C in 5% $CO_2$. One hour post infection 100 U/ml of penicillin/streptomycin (Gibco) was added to kill off extracellular bacteria. RNA was isolated for reverse transcription quantitative PCR (RTqPCR) analysis at 4 h post infection, while supernatants were harvested for ELISA analysis at 20 h post infection.

To transiently permeabilize macrophages infected with wild type or Δ*slo* bacteria, a final concentration of 10 µg/ml digitonin (Sigma) was added with the infections. After a 30 min incubation at 37 °C in 5% $CO_2$, x10 volume of pre-warmed Opti-MEM® was added to dilute out the permeabilizing effect of digitonin. After an additional incubation of 20 min, all medium was removed and replaced with fresh Opti-MEM®.

### Macrophage treatments
To allow for cytosolic access, culture filtrates, molecules or enzymes were diluted in Opti-MEM® and added to macrophages with a final concentration of 10 µg/ml digitonin (Sigma). Cells were incubated for 30 min at 37 °C in 5% $CO_2$ before diluting out the permeabilizing effect of digitonin by addition of x10 volume of pre-warmed Opti-MEM®. After an additional incubation of 20 min at 37 °C in 5% $CO_2$, all medium was removed and replaced with fresh Opti-MEM®. RNA was isolated for RTqPCR analysis at 2.5 h post treatment. Molecules and enzymes were added at the following final concentrations unless otherwise indicated:

Purified recombinant NADase or NADaseG330D (G330D) 0.5 μM[34], c-di-AMP 7.5 μM (InvivoGen), c-di-GMP 7.5 μM (InvivoGen), dsDNA/pTEC15 2.5 ng/ml (Addgene), pI:C 2.5 μg/ml (InvivoGen), LPS 2.5 μg/ml (Sigma). ATP (Thermo Fisher Scientific) was used at final concentrations of 0.1 mM or 1 mM. To degrade c-di-AMP, samples were preincubated (at 32 °C for 60 min) with phosphodiesterase I (Sigma) at a final concentration of 20 U/ml.

### Gene expression analysis

To analyze gene expression in infected and treated macrophages, RTqPCR was performed. RNA was isolated using the RNeasy mini kit (Qiagen) and cDNA was generated according to the manufacturer's instructions with the GoScript Reverse transcription system (Promega). qPCR was performed with SsoFAST EvaGreen Supermix (BioRad) on the CFX384™ Real-time System C1000 Touch™ Thermal Cycler (BioRad) and analyzed using the CFX Maestro software (BioRad). Expression of the housekeeping gene *reep5* was analyzed to enable normalization, and data are represented as a fold-change relative to the UI control. All reagents and primers are listed in Supplementary Table 1.

### Analyses of cytokine secretion and STAT activation

Supernatants from infected macrophages were analyzed by ELISA, using the commercial assays listed in Supplementary Table 1, according to the manufacturer's instructions. Plates were read at 450 nm in the SpectraMAX i3x plate reader (Molecular Devices).

For analysis of STAT1 activation, macrophages were put on ice at the indicated times post infection and lyzed with Nonidet-P40-based lysis buffer (1% NP40 [Sigma]; 150 mM NaCl [Sigma]; 50 mM Tris-base [Sigma], pH 8; 1x Complete EDTA-free protease inhibitor cocktail [Roche]; 1x PhosphoSTOP Easy [Roche]) for 30 min. Cell lysates were separated on 4–15% Mini-PROTEAN® TGX™ Precast Protein Gels (Biorad) and analyzed by Western blot using monoclonal rabbit anti-mouse phospho-STAT1 (p-Tyr701) and polyclonal rabbit anti-mouse STAT1 antibodies (Cell Signaling Technology; see Supplementary Table 1) at a final dilution of 1/2000 for each. For detection, polyclonal goat anti-rabbit IgG conjugated with horseradish peroxidase (Jackson ImmunoResearch; see Supplementary Table 1) was used at a final dilution of 1/5000.

### Analysis of cellular ATP content and lactate dehydrogenase release

Macrophages or cell culture supernatants were collected at indicated time points post infection or treatment and analyzed for ATP content with the Luminescent ATP Detection Assay Kit (Abcam) or LDH release using the colorimetric CytoTox 96 Non-Radioactive Cytotoxicity Assay (Promega), respectively. Both assays were used according to the manufacturer's protocols. Unless otherwise indicated, cytotoxicity was assayed at 4 h post infection, or 2.5 h post treatment of macrophages. LDH release data are presented as percent of max, which was defined by the amount of LDH released upon complete lysis buffer-mediated lysis of cells as described by the manufacturer.

### DNA isolation from whole blood and Sanger sequencing

Genomic DNA was isolated from whole blood samples collected from 73 *S. pyogenes* NSTI patients with the QIAamp DNA Blood Maxi Kit (Qiagen), following the manufacturer's instructions. To identify *STING1* polymorphisms that affect binding of endogenous cGAMP and bacterial c-di-AMP, PCR amplification using the TrueStart Hot Start Taq DNA polymerase (Thermo Scientific) was performed with a set of primers (see Supplementary Table 1) designed to cover the region of *STING1* affected by SNPs rs78233829 and rs1131769, corresponding to amino acid substitutions G230A and R232H, respectively. The Sanger sequencing reaction and analysis were performed using the BigDye™ terminator cycle sequencing kit (Applied Biosystems) and an ABI 3100

capillary sequencing robot, at the Microbial Ecology sequencing unit (Lund University). Raw sequence data were analyzed with Geneious Prime Bioinformatics Software.

### Paleogenomic analysis

We curated a dataset of ~6500 ancient genomes from the Allen Ancient DNA Resource (AADR; V50)[36]. The genotype data of the samples in this public compendium consisted of SNPs from a panel of ~1.24 million markers. Of the SNPs of interest, only rs78233829's tag-SNP (rs7448031) was present in the dataset and was genotyped in 1999 samples (Supplementary Data 1). Minor allele frequencies were calculated for all populations and separately for Europeans and Asians. Modern allele frequencies were obtained from gnomAD and the current variations in STING allele frequencies where from the 1000 genomes project[37].

### Population genetic analysis

We calculated Wright's $F_{ST}$[38] for three continental populations (Africans, Europeans, and Asians), as previously described[39], for *STING1*'s transcripts using the 1000 Genomes data via Ensembl's Variant Effect Predictor (VEP)[40].

### Statistical analysis

Statistical analyses were performed using the software GraphPad Prism version 8. A one-way ANOVA with Dunnett's test or a two-way ANOVA with Tukey's test was used for multiple comparisons, as indicated in figure legends. A $p < 0.05$ was considered significant; *$p < 0.05$, **$p < 0.01$, ***$p < 0.001$, ****$p < 0.0001$. All experiments were repeated at least three times unless otherwise indicated in the figure legend. Analyses of associations between genotype and amputation or septic shock were performed as generalized linear models, using proc genmod in SAS 9.4 (SAS Institute) with phenotype (both amputation and septic shock coded as 0/1) against genotype (homozygous for major allele vs ≥1 minor allele), sex and age. Significance was assessed by likelihood ratio tests. Non-significant factors were deleted at $p \geq 0.05$.

### Reporting summary

Further information on research design is available in the Nature Portfolio Reporting Summary linked to this article.

## Data availability

The authors declare that the data supporting the findings of this study are available within the paper and its supplementary files. Ancient genomes are available at the Allen Ancient DNA Resource[36], and the complete bacterial genomes of the clinical strains are available at the European Nucleotide Archive[41], BioProject PRJNA524111.

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

## Acknowledgements

We thank Arianna Gessa Garcia and Tazin Fahmi for technical assistance, and members of the INFECT study group (www.permedinfect.com) for contributing clinical samples and strains. We are grateful to Gunnar Lindahl for critical reading of the manuscript and Inger Ekström for support with illustrations. These studies were supported by grants from the Swedish Research Council (Dnr: 2018-04777), as well as the foundations of Alfred Österlund, Emil and Wera Cornell, Gunvor and Josef Anér, the Royal Physiographic Society in Lund, and Apotekare Hedberg. Genomic computations were supported by the Swedish Research Council (Dnr: 2020-03485) and enabled by resources provided by the Swedish National Infrastructure for Computing (SNIC) at Lund, partially funded by the Swedish Research Council through grant no. 2018-05973.

## Author contributions

E.M. and F.C. conceived the study. E.M., J.S.B., and C.V. performed all the experiments. J.V. and M.W. purified the NADase proteins. L.R. performed statistical analyses. S.S., M.N., O.H., P.A., M.S., A.N.T., K.H.C., and

M.W. provided critical reagents. E.E. performed the paleogenomic and population genetic analyses. F.C. supervised the project. E.M. and F.C. wrote the manuscript with input from all of the authors.

## Funding

## Competing interests
The authors declare no competing interests.
