## [Peer Review File · Nature Communications]

Interplay between human STING genotype and bacterial NADase activity regulates interindividual disease variabilityREVIEWER COMMENTS

Reviewer #1 (Remarks to the Author):

The authors describe a potentially important new interaction of bacterial virulence determinants with STING.

I would suggest touching on the recent description of the bacterial origin of STING in the introduction - Nature (2020) vol 586:429-433.

Whilst written as a clear contiguous narrative, the authors have used different emm types containing mutations in e.g. SLO, NADase, DacA, GdpP. Some emm types express much higher levels of NADase an SLO compared to others, This generates a concern that different emm types could potentially respond very differently in the assays used here. It would be important to construct an isogenic strain set in a single genetic background (choose from emm1, emm3, emm14 etc) to harmonize the the isogenic strain set (and complement mutants). The mixed emm strain data set presented here could be used to support/validate the work. For an example, there is a huge difference between secreted c-di-AMP between GAS strains HSC5 (Fig. 1b) and M3 (Fig. 1h). How does this allow broad conclusions of the hypothesis across GAS emm types? Figure 1 is a composite of HSC5 and M3 data and Figure 2 contains emm1 data.

In the discussion, the authors highlight differences in association of NADase with invasive disease. The US studies (26,27) would have much less diversity of emm types than the Australian and Worldwide study, where much more emm type variation is documented. The preponderance of emm1 and emm3 (both expressing high NADase) in the US may have skewed the conclusions of these studies - a potential false-positive association?

There appears to be a significant amount of missing methodology. Describe and reference bacterial strains used in the study in the "Bacterial strains" section. Reference previous construction of mutants and describe construction of mutants for this study, with primers used, vectors, methodology etc. Protein precipitation method is missing? Provide primers used for RTqPCR. Describe how much TX-100 used for LDH assay cell lysis? Purification of NADase and G330 methodology?

For bar graph figure presentation, individual measurements of biological replicates are usually now required for data presentation.

Reviewer #2 (Remarks to the Author):

The manuscript by Mover et al examined the role of three *S. pyogenes* virulence factors (SLO, c-di-AMP, and a NADase) on the capacity of bacteria to induce type I interferon and other cytokines/chemokines in mouse bone marrow derived macrophages upon infection. The results show that SLO was necessary to allow c-di-AMP access to the cell cytosol where it activated STING leading to Type 1 IFN. The NADase suppressed this activity, but did not require SLO. The overall hypothesis is that STING activation is protective, but NADase is pathological. Considering that humans have multiple STING alleles, some of which do not respond to c-di-AMP, they hypothesized that the severity of invasive streptococcal infections may be worse in patients harboring STING alleles that do not respond to c-di-AMP. They claim that the human data supported their hypothesis.

1. The use various bacteria strains including one the lacks DacA which I would have thought would be defective. I'd like to know that the bacteria all grow similarly in broth.

2. After 1 hour of macrophage infection, the authors add penicillin and streptomycin to prevent extracellular growth. This was unusual. Investigators often use aminoglycoside antibiotics like streptomycin or gentamicin to kill extracellular bacteria without affecting intracellular bacteria, but I suspect that most of the activity being measured is from extracellular bacteria. Penicillin will likely kill and lyse bacteria resulting in release of bacterial nucleic acids that can act as PAMPS and may contribute to the host response. I'd like to know the rationale for their choice of antibiotics

3. I would like a comment on the pathways that are leading to TNF and CXCL-1.

4. Strain 854 is introduced as a new strain, so it would have been helpful to show that the response was also STING-dependent by using the STING-minus macrophages.

5. The experiment showing that addition of ATP rescued the macrophage responses was impressive. I am curious to see what ATP alone does to cells.

Reviewer #3 (Remarks to the Author):

This is an interesting article, well documented and well written, addressing the important issue of variable outcome of invasive *Streptococcus pyogenes* infection: the authors document an interplay between human STING genotype and bacterial NADase activity in the inflammatory response and clinical outcome. They test the hypothesis that active NADase promotes host-detrimental inflammation by suppressing STING-mediated type I IFN production, which seems to occur via ATP exhaustion.

Some issues could be addressed to strengthen the authors' conclusions:

The sentence on page 8 could be added to the introduction to clarify the activity NADase: "Within cells, active NADase depletes NAD by cleaving the molecule into ADP-ribose and nicotinamide⁸, resulting in ATP deprivation and ultimately cell death^{7,8}."

Suppl Fig.1: The sentences summarizing the results should be more precise and mention the use of KO mice:

"The increased induction of type I IFN by mutants secreting excess amounts of c-di-AMP was completely dependent on STING (Supplementary Fig. 1a) and did not relate to differential cell death (Supplementary Fig. 1b)."

"Production of the neutrophil attractant CXCL1 was inhibited by type I IFN signaling (Supplementary Fig. 1c)"

The sentence on M protein at the end of paragraph on page 5 is asking for more data, or should be moved to the discussion: "Because SLO-dependent translocation is facilitated by the cellular proximity afforded by M protein-mediated adhesion⁶, these results also provide a potential explanation to the role of the streptococcal M protein in type I IFN induction¹²."

Fig.2:

How do the authors explain that 'Infection with the epidemic strains also gave rise to a ~3-fold reduced TNF α response as compared to SF370 (Fig. 2c).'? Does this question the specificity of the link between NADase and type I IFN?

"Lack of NADase (Dnga) destabilizes secreted SLO21, leading to reduced diffusion of c-di-AMP into macrophages, which likely explains why lack of NADase did not affect IFN β output as compared to wild type infection (Fig. 2d).": It is intriguing why absence of NADase does not affect IFN β production/release (as actually seen at mRNA level in Fig.2f). Would it be

possible to measure intracellular c-di-AMP in macrophages to strengthen the explanation proposed by the authors?

“Importantly, the ngaG330D substitution drastically increased secretion of IFN β from infected macrophages (Fig. 2d), demonstrating that NADase activity inhibits STING-mediated type I IFN production.” The implication of STING is not demonstrated here.

“The effect on type I IFN production by the different mutants was not a result of differential secretion of c-di-AMP (Fig. 2g) or cell death (Supplementary Fig. 2) but related to the enzymatic activity of their NADase (Fig. 2h).” It would be important to measure intracellular c-di-AMP in macrophages to strengthen this point.

Is there a correlation between type I IFN production/release and NADase enzymatic activity of the different strains?

Fig.3:

“Type I IFN induction by the TLR3 agonist poly(I:C) was also specifically inhibited by the enzymatically active NADase (Fig. 3a), indicating that NADase acts downstream of STING activation and that its mode-of-action suppresses ifnb expression not only when induced via the STING pathway.”: In general, and here in particular, the data would be strengthened by immunoblot analysis of the pathway downstream of STING /TLR3: how is the phosphorylation of TBK1, IRF3, STAT1, Nf κ Bp65?

“Supplementation with exogenous ATP completely reversed the ability of NADase to inhibit c-di-AMP-induced transcription of ifnb (Fig. 3c).”: How is Tnfa expression affected by NADase +/- ATP? Is the effect selective of IFN β ?

“NADase did not promote LDH release during the time frame of these experiments (Fig.3e), suggesting that its effect on type I IFN production is not a consequence of cell death.” Although there is no overt link to cell death at this time point, could the authors analyze cell/mitochondrial stress, release of mitDNA vs nDNA, and DNA integrity by γ H2AX, to be more specific?

Discussion:

“We identify a new function for the enzymatic activity of bacterial NADase in suppressing c-di-AMP-induced and STING-mediated type I IFN production,”: this seems to be a shortcut, as the authors propose that this happens via ATP exhaustion, which also poses the question of the specificity of the effect for IFN β pathway. More information on the cell stress/death would be informative.

Reviewer #1 (Remarks to the Author):

The authors describe a potentially important new interaction of bacterial virulence determinants with STING.

We appreciate this positive comment regarding our work and the constructive review of our manuscript. All comments are addressed below.

#1. I would suggest touching on the recent description of the bacterial origin of STING in the introduction - Nature (2020) vol 586:429-433.

We thank the Reviewer for pointing out this interesting paper to us. As suggested, we now touch on this recent discovery and include the reference (page 3, ref #13).

#2. Whilst written as a clear contiguous narrative, the authors have used different emm types containing mutations in e.g. SLO, NADase, DacA, GdpP. Some emm types express much higher levels of NADase and SLO compared to others. This generates a concern that different emm types could potentially respond very differently in the assays used here. It would be important to construct an isogenic strain set in a single genetic background (choose from emm1, emm3, emm14 etc) to harmonize the isogenic strain set (and complement mutants). The mixed emm strain data set presented here could be used to support/validate the work. For an example, there is a huge difference between secreted c-di-AMP between GAS strains HSC5 (Fig. 1b) and M3 (Fig. 1h). How does this allow broad conclusions of the hypothesis across GAS emm types? Figure 1 is a composite of HSC5 and M3 data and Figure 2 contains emm1 data.

The Reviewer brings up the point that certain *emm* types might express different levels of NADase activity and SLO, respectively, and that this might impact on how those strains would behave in the assays described in our manuscript. We fully agree that this is likely to be the case. Based on our results one can predict that relatively higher levels of NADase activity would lead to relatively lower type I IFN production due to increased suppression of STING-mediated *ifn β* expression, and that relatively higher levels of SLO would support relatively higher type I IFN responses due to increased diffusion of c-di-AMP. In this context we would like to point out that different strains may also secrete different amounts of c-di-AMP, which can similarly impact on type I IFN production; as pointed out by the Reviewer, we observe a difference between how much c-di-AMP that is secreted from HSC5 (Fig. 1b) compared to M3 (Fig. 1h) and M1 (Fig. 2g) bacteria. Thus, our data suggest that for each bacterial isolate the type I IFN induction would be the result of the relative production of SLO, c-di-AMP and NADase activity. In addition, it is also likely that the type I IFN response could be impacted by, for example, differences in the relative amount of PAMPs that different strains express, which would have an overall impact on the priming of macrophages. However, we fail to see how any of these possibilities constitute a concern regarding any of the interpretations and claims made in our manuscript.

Along the lines of the first part of this comment, the Reviewer continues to ask how our data allow us to draw broad conclusions across *emm* types. We (ref #12) and the group of Pavel Kovarik (ref #11) have previously demonstrated the requirement for STING for two different strains, M5 Manfreda and M1 ISS3348. Here we demonstrate STING-dependence for two additional strains (M14 HSC5 and M1 854). The required and sufficient role for bacteria-derived c-di-AMP is demonstrated for HSC5 in an infection model (Fig. 1b-f) and in a bacteria-free "biochemical" model (Fig. 1g). The role for SLO pore formation is demonstrated in both the M3 system (Fig. 1h-m) and in the M1 854 system (Fig. 2d-g). Based on the Reviewer's comment we have now clarified the point that the required role for pore formation is demonstrated for two separate serotypes (see page 7). The novel role described for NADase is demonstrated in the M1 system using a detailed genetic approach (Fig. 2d-h) as well as by using purified enzymatically active (NADase) and inactive (G330D) proteins (Fig. 3a-e). Thus, we believe that all interpretations and conclusions put forward in our manuscript rest on a very solid experimental foundation. Of note, to date there are over 200 described *emm* types, and it is not feasible to experimentally test all of them, and the generation of an isogenic strain set

(in a single *emm* type) for all the mutations that we examine in the manuscript would not address the questions asked by the Reviewer. Given this situation, the exceptionally ambitious proposition to redo our study with a new set of strains – and to put the current data as a “validation” in the supplement – is in our view beyond the scope of this study.

#3. In the discussion, the authors highlight differences in association of NADase with invasive disease. The US studies (26,27) would have much less diversity of *emm* types than the Australian and Worldwide study, where much more *emm* type variation is documented. The preponderance of *emm1* and *emm3* (both expressing high NADase) in the US may have skewed the conclusions of these studies - a potential false-positive association?

The Reviewer brings up a very interesting point. As described in our manuscript (page 11-12, last section of the discussion) we agree with the Reviewer that the literature on the link between NADase activity and invasive disease paints an ambiguous picture (refs #27-30). We speculated – based on our findings – that differences in the frequencies of minor STING alleles between different human populations might be a factor that contributes to this situation. Indeed, our findings open for a completely novel way (*i.e.* that variability in host genetics put different demands on bacterial NADase activity to produce invasive disease in different populations) to explain the apparent discrepancies between the different clinical studies – a possibility that we, as stated in the discussion, believe warrants further investigation. While our discussion highlights this new and exciting possibility, it does not exclude other potential explanatory models, as the one suggested by the Reviewer. Still, our hypothesis is actually consistent with the preponderance of NADase active serotypes in the Western world (including the US) where the minor STING allele is less common (see Fig. S6a) and the demand for NADase activity (for the development of severe disease) thus expected to be higher. Based on the Reviewer’s comment we have revised this part of the discussion in an attempt to further emphasize the ambiguity of the situation (see page 12).

#4. There appears to be a significant amount of missing methodology. Describe and reference bacterial strains used in the study in the "Bacterial strains" section. Reference previous construction of mutants and describe construction of mutants for this study, with primers used, vectors, methodology etc. Protein precipitation method is missing? Provide primers used for RTqPCR. Describe how much TX-100 used for LDH assay cell lysis? Purification of NADase and G330 methodology?

We thank the Reviewer for requesting clarifications to the methods section, which, we agree, could be improved. Accordingly, it has been revised at several places (see page 12-17). We also wish to refer the Reviewer to Table S1, which includes much of the information asked for.

- All bacterial strains, and a description of their respective genotypes, are described and referenced in Table S1. We agree with the Reviewer that this table could be missed, and have now clarified the reference to Table S1 in the methods section (see page 12).
- All mutants used in our manuscript have been constructed in previous studies, and they have been used and characterized in several previous publications. In Table S1 we provide the appropriate original references.
- All primers used for RTqPCR and genotyping of patients are listed in Table S1. We have now clarified the references to Table S1 (see pages 15 and 16).
- For the LDH assay cell lysis we used the lysis-buffer included in the kit as recommended by the manufacturer. This has now been clarified in the methods section (see page 16).
- We had inadvertently omitted to describe the purification of NADase and NADase-G330D. We now include a brief section (page 13) where we also provide a reference (ref #33) to a more detailed description of the methodology.

#5. For bar graph figure presentation, individual measurements of biological replicates are usually now required for data presentation.

We have revised all bar graphs according to the Reviewer’s suggestion.

Reviewer #2 (Remarks to the Author):

The manuscript by Mover et al examined the role of three *S. pyogenes* virulence factors (SLO, c-di-AMP, and a NADase) on the capacity of bacteria to induce type I interferon and other cytokines/chemokines in mouse bone marrow derived macrophages upon infection. The results show that SLO was necessary to allow c-di-AMP access to the cell cytosol where it activated STING leading to Type 1 IFN. The NADase suppressed this activity, but did not require SLO. The overall hypothesis is that STING activation is protective, but NADase is pathological. Considering that humans have multiple STING alleles, some of which do not respond to c-di-AMP, they hypothesized that the severity of invasive streptococcal infections may be worse in patients harboring STING alleles that do not respond to c-di-AMP. They claim that the human data supported their hypothesis.

We first wish to thank the Reviewer for the constructive review of our manuscript. All comments and questions are addressed below.

#1. The use various bacteria strains including one the lacks DacA which I would have thought would be defective. I'd like to know that the bacteria all grow similarly in broth.

This is an important question, particularly since deletion of the c-di-AMP synthase in Firmicutes may be lethal to the bacteria. However, as previously demonstrated by our collaborator Kyu Hong Cho this is not the case in *S. pyogenes* (ref #20), for unknown reasons. Based on the Reviewer's comment we have now analyzed bacterial growth in THY-broth for all the mutants affected in c-di-AMP metabolism (see Fig. S1). The results demonstrate that the $\Delta dacA$ mutant had a prolonged lag-phase but that it exhibited a similar growth rate as wild type in exponential phase. The other mutants did not exhibit any discernable growth phenotype as compared to wild type. Thus, all strains exhibit similar growth rates in exponential phase when they are harvested for macrophage infections. These results are described on page 4.

#2. After 1 hour of macrophage infection, the authors add penicillin and streptomycin to prevent extracellular growth. This was unusual. Investigators often use aminoglycoside antibiotics like streptomycin or gentamicin to kill extracellular bacteria without affecting intracellular bacteria, but I suspect that most of the activity being measured is from extracellular bacteria. Penicillin will likely kill and lyse bacteria resulting in release of bacterial nucleic acids that can act as PAMPs and may contribute to the host response. I'd like to know the rationale for their choice of antibiotics.

We followed a protocol for streptococcal infection of macrophages that was established in our environment at Lund University many years ago, and that has been used successfully in several publications not only by us (ref #12) but also by others (for example: Carlin AF, Chang YC, Areschough T, *et al.* J Exp Med 2009; Ali SR, Fong JJ, Carlin AF, *et al.* J Exp Med 2014). We agree with the Reviewer that the use of penicillin might cause a generally increased release of intracellular PAMPs (compared to ribosome-targeting antibiotics) that may serve to increase the overall priming of macrophages. However, our findings/claims are based on comparisons between isogenic bacterial strains and knock-out macrophages, all analyzed within this same experimental system. Moreover, the novel function described for NADase is also demonstrated in a bacteria free system (Fig. 3) – as is the role for bacteria-derived c-di-AMP (Fig. 1g) – in which no antibiotics were used. Thus, we are confident that our results regarding the roles of c-di-AMP, SLO and NADase relate specifically to the biological function of these factors, and not to the choice of antibiotics.

#3. I would like a comment on the pathways that are leading to TNF and CXCL-1.

In our manuscript we elucidate the induction and regulation of type I IFN production in *S. pyogenes* infection. TNF α is used as a comparator/control to explore if the effects we observe are selective for IFN β (which they are). Because we find that the production of CXCL-1 is inhibited by type I IFN receptor signaling (Fig. S2c) we included the data presented in Fig. 1f,

since it demonstrates a biological effect downstream of c-di-AMP-induced and STING-dependent type I IFN production that might be of interest to scientists in the field.

Both TNF α and CXCL-1 can be induced via several signals/receptors, and the specific pathways responsible for their production in *S. pyogenes* infection have not, to our knowledge, been well worked out. It would therefore be difficult to (briefly) comment on these pathways without diverting the reader away from the main thrust of the text. Nevertheless, we have previously demonstrated that the production of TNF α in *S. pyogenes* infected macrophages is – unlike the situation for IFN β – dependent on MyD88 (ref #12). Based on the Reviewer's suggestion we now briefly comment on this situation in the results-section (see page 4).

#4. Strain 854 is introduced as a new strain, so it would have been helpful to show that the response was also STING-dependent by using the STING-minus macrophages.

We appreciate the importance of this comment, which is similar to comment #6 by Reviewer #3 (see below). As demonstrated in Fig. 2b and d, the 854 WT strain does essentially not induce any *ifn β* expression, and so it was of particular interest to also examine the increased type I IFN production in *ngaG330D* infection. We have now performed experiments in STING-KO macrophages demonstrating that the induction of type I IFN is indeed completely dependent on STING (see Fig. S3). These results are described on page 7.

#5. The experiment showing that addition of ATP rescued the macrophage responses was impressive. I am curious to see what ATP alone does to cells.

We thank the Reviewer for bringing this important question to our attention. We have now performed experiments indicating that ATP alone does not induce any type I IFN production (see Fig. S5a). These results are described on page 8.

Reviewer #3 (Remarks to the Author):

This is an interesting article, well documented and well written, addressing the important issue of variable outcome of invasive *Streptococcus pyogenes* infection: the authors document an interplay between human STING genotype and bacterial NADase activity in the inflammatory response and clinical outcome. They test the hypothesis that active NADase promotes host-detrimental inflammation by suppressing STING-mediated type I IFN production, which seems to occur via ATP exhaustion.

We thank the Reviewer for these positive remarks regarding our work and for the constructive review of our manuscript. All comments have been addressed to the best of our ability.

Some issues could be addressed to strengthen the authors' conclusions:

#1. The sentence on page 8 could be added to the introduction to clarify the activity NADase: "Within cells, active NADase depletes NAD by cleaving the molecule into ADP-ribose and nicotinamide⁸, resulting in ATP deprivation and ultimately cell death^{7,8}."

We agree and have moved this sentence to the introduction as suggested (see page 3).

#2. Suppl Fig.1: The sentences summarizing the results should be more precise and mention the use of KO mice:

We thank the Reviewer for requesting these clarifications, and have revised the two sentences as suggested:

"The increased induction of type I IFN by mutants secreting excess amounts of c-di-AMP was completely dependent on STING (Supplementary Fig. 1a) and did not relate to differential cell death (Supplementary Fig. 1b)."

The revised text reads (see page 4): *Infection of STING-deficient cells demonstrated that the increased induction of type I IFN by mutants secreting excess amounts of c-di-AMP was completely dependent on STING (Supplementary Fig. 2a) and did not relate to differential cell death (Supplementary Fig. 2b).*

"Production of the neutrophil attractant CXCL1 was inhibited by type I IFN signaling (Supplementary Fig. 1c)"

The revised text reads (see page 4-5): *Production of the neutrophil attractant CXCL1 in infected cells was inhibited by type I IFN signaling as indicated by the increased secretion from type I IFN receptor-deficient macrophages (Supplementary Fig. 2c)*

#3. The sentence on M protein at the end of paragraph on page 5 is asking for more data, or should be moved to the discussion: "Because SLO-dependent translocation is facilitated by the cellular proximity afforded by M protein-mediated adhesion⁶, these results also provide a potential explanation to the role of the streptococcal M protein in type I IFN induction¹²."

We agree and have moved this sentence to the discussion as suggested (see page 11, middle section).

Fig.2:

#4. How do the authors explain that 'Infection with the epidemic strains also gave rise to a ~3-fold reduced TNF α response as compared to SF370 (Fig. 2c).'? Does this question the specificity of the link between NADase and type I IFN?

We thank the Reviewer for this question, which has prompted clarifications in our manuscript. In Fig. 2a-c we analyzed one pre-epidemic (SF370) and two epidemic (5448 and 854) strains. While we cannot explain the ~3-fold reduced TNF α output from macrophages infected with the epidemic strains, we do know that this result does not question the specificity of our findings. Importantly, these strains are not isogenic, and so to investigate the role for NADase in regulating the type I IFN response we employed a set of well characterized isogenic mutants in the 854 background (Fig. 2d-h). These analyses demonstrated that the enzymatic activity

of NADase regulates type I IFN production specifically (compare **Fig. 2d** with **e**). Unlike the situation for IFN β (**Fig. 2d**), TNF α production is apparently similarly affected by all mutations in the NADase and SLO encoding operon (**Fig. 2e**). While these results suggest that TNF α production is somehow affected by the operon it also demonstrates that it must be so via a mechanism that is distinct from the one suppressing type I IFN production. Thus, our findings with non-isogenic strains presented in **Fig. 2c** do not question the specificity of the link between the enzymatic activity of NADase and type I IFN production. Since the regulation of TNF α production is outside the scope of the current study we simply state, in the manuscript, that (see **page 7**): *The TNF α output from infected cells was not specifically regulated by NADase activity (Fig. 2e), indicating that the enzymatic activity of NADase selectively inhibits production of type I IFN.*

#5. “Lack of NADase (Dnga) destabilizes secreted SLO21, leading to reduced diffusion of c-di-AMP into macrophages, which likely explains why lack of NADase did not affect IFN β output as compared to wild type infection (Fig. 2d).”: It is intriguing why absence of NADase does not affect IFN β production/release (as actually seen at mRNA level in Fig.2f). Would it be possible to measure intracellular c-di-AMP in macrophages to strengthen the explanation proposed by the authors?

Initially we were puzzled by the finding that absence of NADase did not affect IFN β production as measured by ELISA at 20 hpi (**Fig. 2d**) while there was a limited but significant effect as measured by RTqPCR at 4 hpi (**Fig. 2f**). We know from earlier work that lack of NADase causes destabilization of SLO (**ref #22**). Thus, deletion of the *nga* gene is expected to significantly reduce c-di-AMP-diffusion into the macrophage cytosol and therefore significantly reduce the induction of IFN β production. We also know that the production of IFN β in infected macrophages peaks early and that the concentration of IFN β protein in the cell culture supernatant subsequently declines (see Fig. 4c in **ref #12**). Thus, our findings are likely explained by Δnga bacteria causing a limited production of IFN β (**Fig. 2f**) that is not detectable at the protein level at 20 hpi (**Fig. 2d**).

We agree with the Reviewer that it would be of interest to directly measure the level of bacteria-derived c-di-AMP in the cytosol of infected macrophages. To this end we have purified the cytosolic compartment of infected cells using a protocol developed and used for two previous projects in our group (**ref #12** and Lienard J, *et al.* PNAS 2020). Unfortunately, our analyses demonstrated that the cytosolic levels of c-di-AMP in infected cells are too low to be reliably analyzed by the c-di-AMP ELISA kit. Despite significant efforts to overcome this problem we have not been able to do so, precluding our ability to perform the suggested analysis.

#6. “Importantly, the *ngaG330D* substitution drastically increased secretion of IFN β from infected macrophages (Fig. 2d), demonstrating that NADase activity inhibits STING-mediated type I IFN production.” The implication of STING is not demonstrated here.

We thank the Reviewer for bringing up this key point. We fully agree that it was of importance to confirm that the elevated level of IFN β production in *ngaG330D* infected macrophages is STING-dependent. As described in the response to comment #4 by Reviewer #2 (see above), we have now performed the appropriate experiments demonstrating this to be the case (**Fig. S3**). These new results are described on **page 7**.

#7. “The effect on type I IFN production by the different mutants was not a result of differential secretion of c-di-AMP (Fig. 2g) or cell death (Supplementary Fig. 2) but related to the enzymatic activity of their NADase (Fig. 2h).” It would be important to measure intracellular c-di-AMP in macrophages to strengthen this point. Is there a correlation between type I IFN production/release and NADase enzymatic activity of the different strains?

As described in our response to comment #5 above, we agree that it would have been of interest to measure the level of bacteria-derived c-di-AMP in the cytosol of infected

macrophages. Unfortunately, we have not been able to do so, precluding our ability to perform the analysis.

The Reviewer's question regarding the correlation between type I IFN production/release and NADase enzymatic activity of the different strains made us realize that we needed to clarify the interpretation we made (which was poorly described). Based on the data presented in **Fig. 2** the revised text (see **page 7**, last sentence) now reads: *The ability of the bacterial strains to suppress STING-mediated type I IFN production was not a result of differential secretion of c-di-AMP (Fig. 2g) or cell death (Supplementary Fig. 4) but related specifically to their ability to translocate an enzymatically active NADase (Fig. 2d-f and h).* We hope that we have hereby clarified the situation.

Fig.3:

#8. "Type I IFN induction by the TLR3 agonist poly(I:C) was also specifically inhibited by the enzymatically active NADase (Fig. 3a), indicating that NADase acts downstream of STING activation and that its mode-of-action suppresses *ifnb* expression not only when induced via the STING pathway.": In general, and here in particular, the data would be strengthened by immunoblot analysis of the pathway downstream of STING /TLR3: how is the phosphorylation of TBK1, IRF3, STAT1, NFkBp65?

We agree that such analyses would be of interest. To us it would be of particular interest to investigate the activation status of the factors downstream of STING, *i.e.* TBK1 and IRF3. While information regarding their activation status would not stand to impact on any of the interpretations or claims made in the current manuscript, it might guide future studies aimed at elucidating the mechanism by which NADase suppresses *ifnb* expression. We have therefore made significant efforts to perform the suggested experiments. However, despite testing several commercially available antibodies against phosphorylated and total TBK1, and IRF3, we have not been able to visualize these proteins by Western blot, for unknown reasons.

#9. "Supplementation with exogenous ATP completely reversed the ability of NADase to inhibit c-di-AMP-induced transcription of *ifnb* (Fig. 3c).": How is *Tnfa* expression affected by NADase +/- ATP? Is the effect selective of IFN1?

To address this question we have now analyzed *TNFα* expression and find that the addition of c-di-AMP does not induce significant expression of *TNFα* (**Fig. 3C**), precluding our ability to test if NADase and ATP affect *TNFα* in this experimental setting. However, our analysis of infected macrophages provides an answer to the question of specificity. Indeed, in contrast to IFNβ production (**Fig. 2d**) the production of *TNFα* in infected cells is not specifically affected by the enzymatic activity of NADase (**Fig. 2e**), demonstrating that the enzymatic activity of NADase selectively regulates type I IFN production via a distinct mechanism.

#10. "NADase did not promote LDH release during the time frame of these experiments (Fig.3e), suggesting that its effect on type I IFN production is not a consequence of cell death." Although there is no overt link to cell death at this time point, could the authors analyze cell/mitochondrial stress, release of mitDNA vs nDNA, and DNA integrity by pgH2AX, to be more specific?

We have previously shown that *S. pyogenes* induces type I IFN production in an cGAS-independent (cGAS-KO has no phenotype) but STING-dependent manner (see Fig. 5b in **ref #12**). We also showed that infection with *S. pyogenes* does not lead to cytosolic release of any DNA – neither mitochondrial, nuclear nor bacterial (see Fig. 6a in **ref #12**). As control we similarly analyzed infection with mycobacteria, which induced type I IFN production in a cGAS- and STING-dependent manner (see Fig. 6c in **ref #12**) and which caused the release of both mitochondrial and nuclear DNA into the cytosol of infected cells (see Fig. 6b in **ref #12**). Since cGAS is the only DNA sensor capable of producing an endogenous cyclic dinucleotide (cGAMP) to activate STING these findings essentially excludes the possibility of DNA – as well as cellular stress leading to cytosolic release of DNA – playing a role in the induction of type I IFN production in *S. pyogenes* infection. In the current manuscript we provide data from

both infection (**Fig. S1b, Fig. S2**) and the use of purified NADase protein (**Fig. 3e**) that argues against a role for toxicity in the ability of NADase to suppress type I IFN production. Because all available data argues against a role for cellular stress and DNA in our system we have opted not to pursue the suggested analyses. Please also read our response to comment #11 below.

Discussion:

#11. “We identify a new function for the enzymatic activity of bacterial NADase in suppressing cdi-AMP-induced and STING-mediated type I IFN production,”: this seems to be a shortcut, as the authors propose that this happens via ATP exhaustion, which also poses the question of the specificity of the effect for IFN1 pathway. More information on the cell stress/death would be informative.

Using both genetic and biochemical approaches our data demonstrates that NADase suppresses c-di-AMP-induced and STING-mediated type I IFN production in *S. pyogenes* infection (which is not questioned by any of the Reviewers). Our data also suggest that this effect is selective since TNF α is not similarly affected by the enzymatic activity of NADase (see, for example, **Fig. 2d and e**). A big future question for us – which is at the core of this comment by the Reviewer – is to elucidate the downstream events (*i.e.* molecular mechanism) by which NADase regulates type I IFN production. While our results indicate a role for ATP exhaustion, the direct mechanism remains a completely open question. Based on the Reviewer’s comment we have revised the discussion to clarify that the mechanism by which NADase suppresses type I IFN production remains to be elucidated (see **page 11**), as to avoid any impression of us taking shortcuts. However, our data from both infection (**Fig. S1b, Fig. S2**) and the use of purified NADase protein (**Fig. 3e**) argues against a role for toxicity, and therefore do not prompt detailed studies along that avenue. Rather, and very briefly, preliminary data from our ongoing efforts to elucidate the mechanistic basis for our herein described findings suggest a possible role for regulation of acetylation/deacetylation of a yet unidentified target protein (which we hypothesize might be TBK1 or IRF3). However, these studies are in their infancy and the elucidation of the mechanisms is well beyond the scope of the current manuscript.

REVIEWER COMMENTS

Reviewer #1 (Remarks to the Author):

The authors have not fulfilled Koch's molecular postulates in this investigation.

The authors have used different emm types containing mutations in e.g. SLO, NADase, DacA, GdpP. Some emm types express much higher levels of NADase an SLO compared to others, This generates a concern that different emm types could potentially respond very differently in the assays used here. It is important to construct an isogenic strain set in a single genetic background (choose from emm1, emm3, emm14 etc) to harmonize the the isogenic strain set and complement mutants).

Without fulfilling Koch's molecular postulates, this investigation is flawed.

Reviewer #2 (Remarks to the Author):

Dear Editor: I am satisfied by the authors responses to my review.

Reviewer #3 (Remarks to the Author):

The authors addressed all questions raised during review in a satisfactory way.

Reviewer #1 (Remarks to the Author):

The authors have not fulfilled Koch's molecular postulates in this investigation. The authors have used different emm types containing mutations in e.g. SLO, NADase, DacA, GdpP. Some emm types express much higher levels of NADase an SLO compared to others, This generates a concern that different emm types could potentially respond very differently in the assays used here. It is important to construct an isogenic strain set in a single genetic background (choose from emm1, emm3, emm14 etc) to harmonize the the isogenic strain set and complement mutants). Without fulfilling Koch's molecular postulates, this investigation is flawed.

We thank the Reviewer for the thorough review, and we were happy to see that all but one comments are considered resolved based on our previous revision. The current comment is a continuation of comment #2 from the previous round of review – focused on how our findings would “allow broad conclusions of the hypothesis across GAS emm types” given that the expression of NADase, SLO, DacA and GdpP may vary between strains. As described below we have now addressed this valid question experimentally.

SLO and NADase are both transcribed under the same promoter and their expression levels therefore co-vary. In our study the relevance of DacA and GdpP lies in that they both regulate the amount of c-di-AMP that is secreted from the bacteria (**Fig. 1b**). We fully agree with the Reviewer that it would be of interest to investigate the impact of the levels of the two key bacterial variables – *i.e.* c-di-AMP secretion (causing activation of STING) and NADase activity (causing suppression of STING-mediated type I IFN production) – across infections with different strains. Analyses of the 53 clinical strains (which include 13 different *emm*-types, **see Supplementary Fig. 7**) demonstrate that the level of NADase activity exhibits a significant positive correlation with the development of septic shock in patients (**Fig. 4f**). Prompted by the Reviewer's comment we have now performed extensive experiments with these strains to similarly analyze the impact of differential c-di-AMP secretion. The new results are shown in **Supplementary Fig. 10** and described on **page 10** in the revised manuscript. Importantly, in contrast to NADase activity (**Fig. 4f**), the amount of secreted c-di-AMP did not correlate with the development of septic shock (**Supplementary Fig. 10**). Thus, analyses of 53 clinically relevant strains suggest that NADase activity, but not the amount of secreted c-di-AMP (*i.e.* not DacA and GdpP), is the biologically relevant variable across different strains and *emm*-types in invasive human infection. We thank the Reviewer for the comment, which we believe has served to improve our manuscript.

Of note, as detailed in our previous response we employed comprehensive genetic and biochemical approaches to demonstrate the required and the sufficient role for secreted c-di-AMP to activate STING, as well as for NADase activity to suppress STING-mediated type I IFN production. Thus, our interpretations rest on an experimental basis that goes beyond “molecular Koch's postulates”, which do not address the key issue of sufficiency. Moreover, our findings are also based on the analyses of a large number of clinical strains and patients, demonstrating the interaction between NADase activity and STING genotype in human infection. In our view, the request that we should generate a completely new set of strains and then redo our study is therefore beyond the scope of the current investigation, and it would not (unlike the new results provided in **Supplementary Fig. 10**) affect our ability to draw conclusions across strains and *emm*-types.

Reviewer #2 (Remarks to the Author):

Dear Editor: I am satisfied by the authors responses to my review.

We were happy to see that this Reviewer was satisfied with our revised manuscript, and we thank the Reviewer for contributing to improving our manuscript.

Reviewer #3 (Remarks to the Author):

The authors addressed all questions raised during review in a satisfactory way.

We were happy to see that this Reviewer was satisfied with our revised manuscript, and we thank the Reviewer for contributing to improving our manuscript.

REVIEWERS' COMMENTS

Reviewer #1 (Remarks to the Author):

The authors have made useful improvements to the manuscript. I would suggest that genetic experiments of this type should include fulfilling molecular Koch's postulates. There is a rationale, as set out in the original publication by Stan Falkow, for doing so.

Note that NADase and SLO expression are linked, and thus ant correlation of NADase activity and virulence would also show similar correlation with SLO expression.

Reviewer #1 (Remarks to the Author):

The authors have made useful improvements to the manuscript. I would suggest that genetic experiments of this type should include fulfilling molecular Koch's postulates. There is a rationale, as set out in the original publication by Stan Falkow, for doing so.

We thank the Reviewer for acknowledging the improvements to our manuscript. Based on the current comment we have revised our manuscript (see page 11; changes highlighted in yellow) to discuss that our study did not include complementation (to fulfill molecular Koch's postulates) of bacterial mutants. We have also included the reference (ref#27) to Stanley Falkow's 1988-paper where he coined, and described the rationale for, "molecular Koch's postulates".

(page 11): *"While our bacterial mutants were not complemented to formally exclude contributions from potential secondary mutations²⁷, interpretations based on macrophage infection experiments were validated by biochemical approaches and supported by clinical data."*

Note that NADase and SLO expression are linked, and thus ant correlation of NADase activity and virulence would also show similar correlation with SLO expression.

It is correct that the mRNA expression levels of NADase (*nga*) and SLO (*slo*) are expected to co-vary since they are expressed under the same promoter. This is indicated in our introduction (including the appropriate references [ref#2-4]) on page 2. It should be noted, however, that NADase activity is a product of both the expression level and of SNPs affecting the enzymatic activity of the protein, and is thus not merely a product of gene expression. Still, we agree with the Reviewer and have revised our manuscript to highlight the link between *nga* and *slo* expression (see page 10; changes highlighted in yellow).

(page 10): *"However, individuals infected with strains expressing relatively higher NADase activity were more likely to develop this systemic inflammatory condition (Fig. 4f). The amount of secreted c-di-AMP did not similarly correlate with septic shock (Supplementary Fig. 10). Though the expression levels of *slo* and *nga* co-vary³, these findings identify NADase activity, and not c-di-AMP secretion, as the key bacterial variable across different strains and emm-types."*

Finally, we would like to take this opportunity to thank the Reviewer for the thorough review and for contributing to improving our manuscript.